# The Ku subunit of telomerase binds Sir4 to recruit telomerase to lengthen telomeres in *S. cerevisiae*

**Evan P Hass, David C Zappulla\***

Department of Biology, Johns Hopkins University, Baltimore, United States

**Abstract** In *Saccharomyces cerevisiae* and in humans, the telomerase RNA subunit is bound by Ku, a ring-shaped protein heterodimer best known for its function in DNA repair. Ku binding to yeast telomerase RNA promotes telomere lengthening and telomerase recruitment to telomeres, but how this is achieved remains unknown. Using telomere-length analysis and chromatin immunoprecipitation, we show that Sir4 – a previously identified Ku-binding protein that is a component of telomeric silent chromatin – is required for Ku-mediated telomere lengthening and telomerase recruitment. We also find that specifically tethering Sir4 directly to Ku-binding-defective telomerase RNA restores otherwise-shortened telomeres to wild-type length. These findings suggest that Sir4 is the telomere-bound target of Ku-mediated telomerase recruitment and provide one mechanism for how the Sir4-competing Rif1 and Rif2 proteins negatively regulate telomere length in yeast.

## Introduction

The ends of linear eukaryotic chromosomes are protected by telomeres, which contain DNA repeats that buffer against shortening caused by the end-replication problem. In most eukaryotes, telomeres are lengthened by the enzyme telomerase (*Greider and Blackburn, 1985*). Telomerase is a multi-subunit ribonucleoprotein (RNP) complex, containing the telomerase reverse transcriptase (TERT) and the telomerase RNA, which contains the template for telomeric repeat synthesis by TERT (*Shippen-Lentz and Blackburn, 1990*). The telomerase RNA is more than just a template, however, as it has conserved structures in its core that are required for catalytic activity (*Bhattacharyya and Blackburn, 1994*; *Tzfati et al., 2000*; *Lin et al., 2004*; *Mefford et al., 2013*; *Niederer and Zappulla, 2015*) and serves as a scaffold for assembling the telomerase RNP holoenzyme (*Zappulla and Cech, 2004*; *Zappulla et al., 2011*; *Lebo and Zappulla, 2012*; *Lebo et al., 2015*).

Telomeric DNA and neighboring subtelomeric regions are often packaged into heterochromatin. In the yeast *Saccharomyces cerevisiae*, telomeric silent chromatin is largely composed of the histone deacetylase Sir2 and structural components, Sir3 and Sir4. Complexes of these three proteins are recruited to telomeric DNA in part by the DNA-binding protein Rap1, which interacts with Sir3 and Sir4 (*Moretti et al., 1994*). Sir2/3/4 complexes then associate with hypoacetylated H3 and H4 tails from telomeric into subtelomeric regions (*Luo et al., 2002*) and can cause silencing of telomere-proximal genes (*Gottschling et al., 1990*). In addition to the Sir2/3/4 complex, telomeric silencing also requires the Ku heterodimer (*Boulton and Jackson, 1998*), a highly conserved DNA end-binding complex of the proteins Ku70 and Ku80 (Yku70 and Yku80 in yeast) well known for its function in non-homologous end-joining. Ku binds telomeres (*Martin et al., 1999*) and has been found to interact with Sir4 in two-hybrid screens as well as by co-immunoprecipitation (*Tsukamoto et al., 1997*; *Roy et al., 2004*). Ku also protects the telomeric 5′ end from resection (*Gravel et al., 1998*; *Polotnianka et al., 1998*; *Bonetti et al., 2010*) and is a subunit of the telomerase holoenzyme.

\*For correspondence:
zappulla@jhu.edu

**Competing interests:** The authors declare that no competing interests exist.

**eLife digest** Inside a cell's nucleus, DNA is packaged into structures called chromosomes. The ends of every chromosome are capped by repeating sequences of DNA known as telomeres, which protect the chromosomes from damage. Every time a cell divides, the telomeres shorten. If telomere length falls below a critical level, the cell can die or enter a state in which it can no longer divide.

During cell division, an enzyme called telomerase normally restores telomeres to their original length. Telomerase is made up of several proteins and an RNA molecule. In yeast and humans, a protein called Ku is one part of the telomerase enzyme. Ku binds to the RNA subunit of telomerase and helps the enzyme find and interact with the telomeres. Previous research has shown that Ku is unable to work alone to recruit telomerase to the chromosome.

A protein called Sir4 binds to telomeres and cells lacking it have short telomeres, but the reason behind this was not known. Hass and Zappulla confirmed previous reports that Ku binds to Sir4 using a biochemical approach. Additional experiments provided genetic evidence that this binding interaction is important for telomerase to lengthen telomeres appropriately.

Cells in which the RNA subunit of telomerase is unable to bind effectively to Ku have short telomeres. Hass and Zappulla directly tethered Sir4 to this defective RNA and found this restored the shortened telomeres to a normal length, indicating that Sir4 normally binds Ku to recruit telomerase. Discovering this mode of recruitment also helps to explain how two other telomeric proteins (Rif1 and 2) limit telomere lengthening; they compete with Ku-Sir4 recruitment to form a length-regulating system.

Taken together, Hass and Zappulla's results provide strong evidence that Sir4 cooperates with Ku to control the lengthening of chromosome ends. Future research will hopefully reveal the precise space and time requirements for this telomerase-controlling system in yeast. Additionally, because Ku has been reported to be a subunit of human telomerase, future studies could also explore whether human cells use a similar strategy.

Ku binds telomerase RNA both in yeast (*Peterson et al., 2001*; *Stellwagen et al., 2003*; *Dalby et al., 2013*) and in humans (*Ting et al., 2005*). In *S. cerevisiae*, Ku binds to the tip of a 74-nt hairpin in the 1157-nt telomerase RNA, TLC1 (*Figure 1*) (*Peterson et al., 2001*; *Stellwagen et al., 2003*; *Dandjinou et al., 2004*; *Zappulla and Cech, 2004*; *Dalby et al., 2013*). In cells where the tip of this hairpin is deleted (*tlc1Δ48*), telomeres shorten by ~70 base pairs (*Peterson et al., 2001*; *Stellwagen et al., 2003*; *Zappulla et al., 2011*). This defect can be mostly rescued by inserting a Ku-binding hairpin at other locations within the mutant tlc1Δ48 RNA, whereas inserting additional Ku-binding hairpins into wild-type TLC1 causes progressive telomere hyper-elongation (*Zappulla et al., 2011*). Lack of Ku binding to TLC1 has also been reported to reduce nuclear localization of TLC1 (*Gallardo et al., 2008*) and recruitment of telomerase to telomeres (*Fisher et al., 2004*). Also important for telomerase recruitment to telomeres is the protein Est1. Est1 was the first telomerase subunit identified (*Lundblad and Szostak, 1989*) and is required for recruiting telomerase to telomeres through an interaction with the single-stranded telomeric DNA-binding protein Cdc13 (*Evans and Lundblad, 1999*; *Qi and Zakian, 2000*). In contrast to Est1, the mechanism by which Ku recruits telomerase to telomeres has yet to be elucidated.

An initial working model for Ku-mediated telomerase recruitment to telomeres was that TLC1-bound Ku simply recruits telomerase to telomeres by also binding telomeric DNA (*Peterson et al., 2001*; *Fisher and Zakian, 2005*). However, this model has been largely discounted by in vitro binding experiments showing that purified Ku cannot bind DNA and RNA concurrently (*Pfingsten et al., 2012*). It therefore seemed likely to us that Ku recruits telomerase to telomeres by interacting with a telomere-associated protein. Such a protein must bind Ku and associate with telomeres. Knowing that Ku also plays a role in the formation of telomeric silent chromatin, we chose to investigate its binding partner in this process, the protein Sir4, as a possible candidate. Sir4 associates with telomeres, and as mentioned above, an interaction between Sir4 and Ku has been reported previously (*Tsukamoto et al., 1997*; *Roy et al., 2004*). Additionally, *sir4Δ* cells have been shown to have shortened telomeres (*Palladino et al., 1993*; *Askree et al., 2004*; *Gatbonton et al., 2006*), although the cause of this phenotype has remained apparently unexplored.

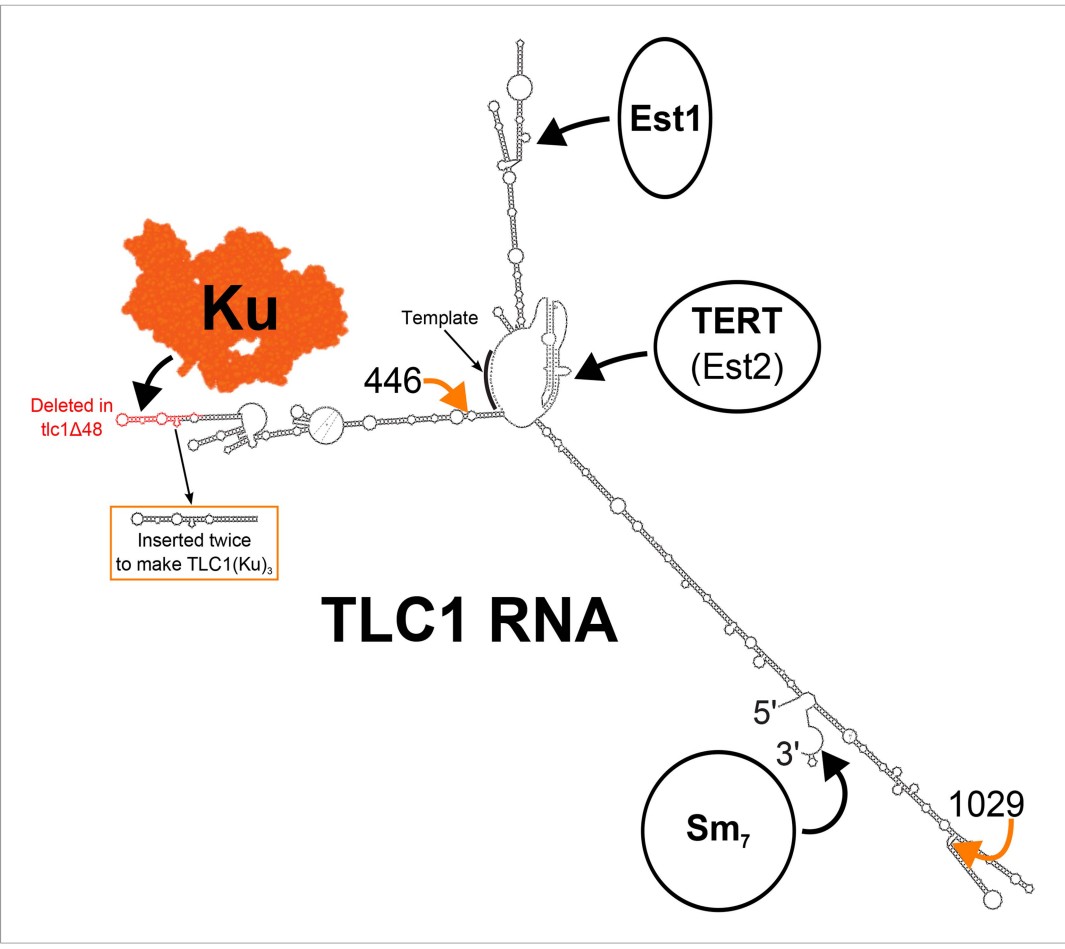

**Figure 1**. Secondary structure model of yeast telomerase RNA. The 48 nucleotides deleted in the *tlc1Δ48* allele are highlighted in red. The 74-nucleotide hairpin shown in the orange box was inserted at positions 446 and 1029 (indicated by the orange arrows) to create *TLC1(Ku)₃*. The TLC1 secondary structure shown is based on previously published models of the core (*Niederer and Zappulla, 2015*) and of the arms (*Dandjinou et al., 2004*; *Zappulla and Cech, 2004*), while the Ku crystal structure shown is that of the human Ku70/80 complex (*Walker et al., 2001*).

Here, we provide genetic evidence suggesting that *SIR4* and TLC1-bound Ku promote telomere lengthening through the same pathway and that *SIR4* is required for Ku-mediated telomere lengthening. In contrast, the negative regulators of telomerase, Rif1 and Rif2, which compete with Sir3 and Sir4 for binding to Rap1 (*Moretti et al., 1994*; *Wotton and Shore, 1997*), appear to inhibit Ku-mediated telomere lengthening. By measuring telomerase recruitment to telomeres by chromatin immunoprecipitation (ChIP), we find that a TLC1 RNA containing three Ku-binding sites, TLC1(Ku)₃, causes increased telomerase recruitment in wild-type cells. Furthermore, *sir4Δ* cells display a defect in telomerase recruitment indistinguishable from that of *tlc1Δ48* cells, even when expressing TLC1(Ku)₃. Finally, we show that tethering Sir4 directly to tlcΔ48 RNA restores telomeres to wild-type length, while tethering Sir3 to tlc1Δ48 does not. Together, these results suggest that Ku recruits telomerase to telomeres through its interaction with Sir4 and that this recruitment pathway is counterbalanced by Rif1 and Rif2.

## Results

### *Ku*, *SIR4*, and the *TLC1* Ku-binding site promote telomere lengthening through the same pathway

Although the exact mechanism of Ku-mediated telomerase recruitment remains unclear, a simple model is that Ku recruits telomerase to telomeres by binding a telomere-bound protein. The telomeric silent chromatin protein Sir4 is an attractive candidate for playing this role, since it has been shown to bind Ku

and because *sir4Δ* cells have telomeres 50–150 bp shorter than wild type (*Palladino et al., 1993*; *Gatbonton et al., 2006*), a phenotype similar to the ~70-bp reduction seen in *tlc1Δ48* cells (*Peterson et al., 2001*; *Stellwagen et al., 2003*; *Zappulla et al., 2011*). As a first test of the hypothesis that *SIR4* is involved in Ku's function as a telomerase subunit, we accurately measured the length of telomeres in *sir4Δ* cells and *tlc1Δ48* cells, as well as *sir4Δ tlc1Δ48* double mutants. We found that telomeres in *tlc1Δ48* cells were 85 ± 23 bp shorter than wild type, while those in *sir4Δ* cells were 53 ± 13 bp shorter than wild type (*Figure 2A*, *Table 1*). When these two mutations were combined to make

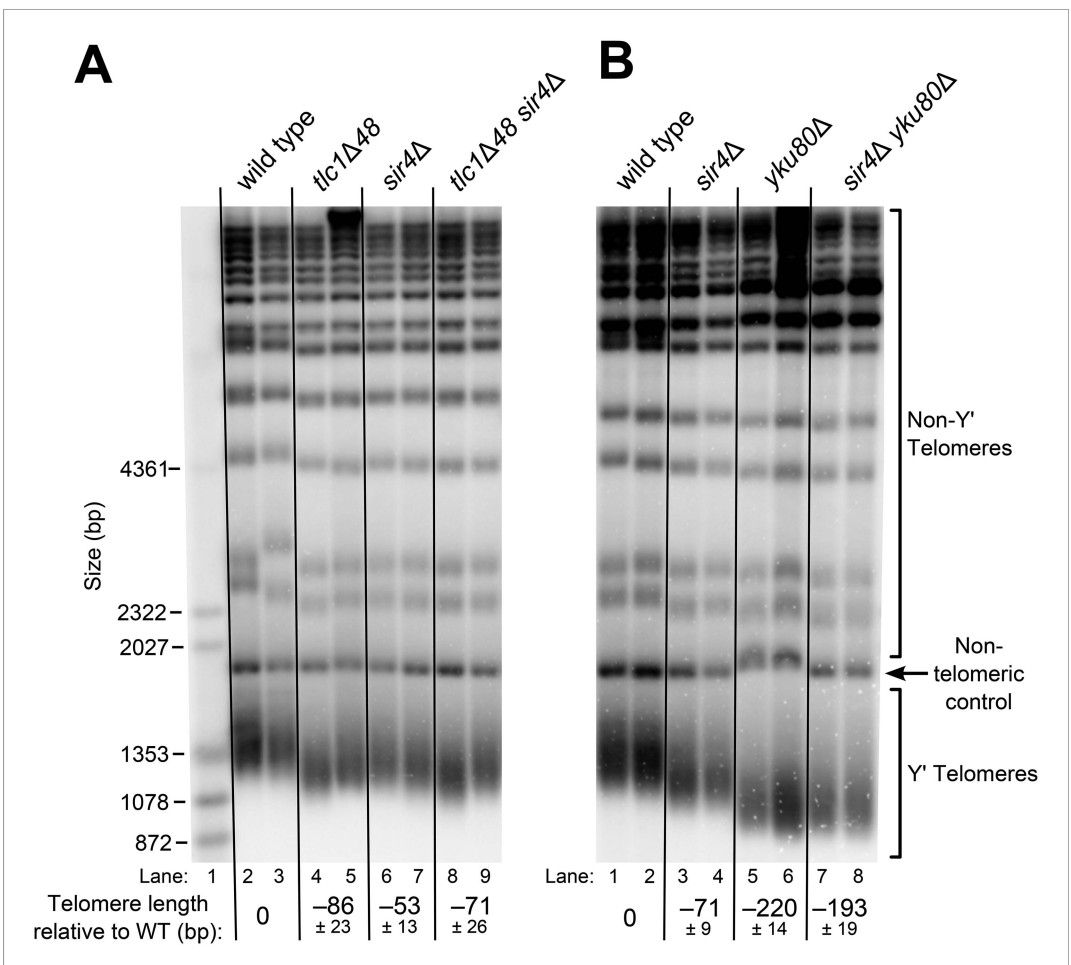

**Figure 2**. *SIR4,* Ku, and the Ku-binding site in TLC1 are in the same telomere-lengthening pathway. (**A**) Deleting *SIR4* in *tlc1Δ48* cells does not cause further telomere shortening. A *tlc1Δ pTLC1-URA3* strain and an isogenic *sir4Δ* strain were transformed with *CEN* plasmids expressing either TLC1 or tlc1Δ48, and then the *pTLC1-URA3* cover plasmid was shuffled out. The cells were serially re-streaked five times, and genomic DNA was isolated and analyzed by Southern blotting. The Southern blot was probed for telomeric sequence and for a 1621-bp non-telomeric XhoI restriction fragment from chromosome IV ('non-telomeric control') used as a relative-mobility control. Pairs of lanes represent independent transformants. Changes in telomere length were quantitated using the Y' telomere bands as described in 'Materials and methods'. Telomere lengths calculated from the two sets of replicates shown were averaged with telomere lengths from four other sets of replicate samples from similar experiments to give the numbers shown, ± standard deviation. The numbers shown here are the same as those in *Table 1*. (**B**) Deleting *SIR4* in *yku80Δ* cells does not cause further telomere shortening. A *SIR4/sir4Δ YKU80/yku80Δ* diploid strain was sporulated, and tetrads were dissected. The haploid spores of a tetratype tetrad were serially re-streaked three times on plates to equilibrate telomere length before Southern blot analysis. The pairs of lanes on the blot shown are different colonies from streak-outs of the haploid spores. Telomere lengths calculated from the two sets of replicates shown were averaged with telomere lengths from a third set of replicate samples to give the numbers shown, ± the standard deviation.

**Table 1.** Average Y' telomere length in *sir*Δ cells containing TLC1, tlc1Δ48, or TLC1(Ku)$_3$

| SIR Genotype | TLC1 Genotype | | |
| --- | --- | --- | --- |
| | TLC1 | tlc1Δ48 | TLC1(Ku)$_3$ |
| SIR | 0 | −86 ± 23† | Dysregulated* |
| sir4Δ | −53 ± 13† | −71 ± 26† | −148 ± 36‡ |
| sir2Δ§ | −41 ± 16 | −50 ± 74 | −71 ± 26 |
| sir3Δ§ | −51 ± 20 | −84 ± 14 | −123 ± 2 |

The weighted-average mobility of the Y' telomeric restriction fragments was calculated as described in the 'Materials and methods'. The numbers shown are averages of multiple biological-replicate samples ± standard deviation.
*Y' telomere length was not quantified in this condition because signal from Y' telomere restriction fragments overlapped with that from the non-telomeric control fragment.
†n = 6.
‡n = 4.
§n = 2.

a double-mutant strain, telomeres were 71 ± 26 bp shorter than wild type, a telomere length defect very similar to that of the *tlc1Δ48* single-mutant (p = 0.31). This genetic epistasis suggests that *SIR4* promotes telomere lengthening in the same pathway as TLC1-bound Ku.

To test if this result is, in fact, indicative of related function in telomere-length maintenance between *TLC1*, *Ku*, and *SIR4,* and not simply between *TLC1* and *SIR4*, we performed a similar genetic epistasis experiment with *sir4Δ* and *yku80Δ* mutants. Similar to what has been reported previously, we observed that *yku80Δ* cells supported telomeres 220 ± 14 bp shorter than wild type (*Figure 2B*) (*Gravel et al., 1998*; *Askree et al., 2004*; *Gatbonton et al., 2006*). However, while deleting *SIR4* resulted in a ~70-bp telomere-length defect in a wild-type background, it appeared to have little effect on telomere length in a *yku80Δ* background; telomeres in *sir4Δ yku80Δ* cells were 193 ± 19 bp shorter than wild type. None of the strains in these experiments senesced (data not shown), and telomeres have been reported to shorten by as much as ~260 bp in other mutants without causing senescence (*Lebo and Zappulla, 2012*). Thus, the lack of further telomere shortening in the *sir4Δ yku80Δ* double-mutant strain relative to *yku80Δ* is not explained by telomeres already being the shortest-possible length supporting cell growth. These findings suggest that Ku is involved in the same telomere length-maintenance pathway as *SIR4*.

## Telomere hyper-lengthening by telomerase RNA with extra Ku-binding sites is *SIR4*-dependent

Inserting an extra Ku-binding hairpin into TLC1 causes progressive telomere hyper-lengthening (*Zappulla et al., 2011*). Furthermore, we have generated a telomerase RNA, TLC1(Ku)$_3$, that contains extra Ku-binding hairpins inserted at positions 446 and 1029. This TLC1(Ku)$_3$ telomerase RNA accumulates to essentially the same level (93 ± 9%) as wild-type TLC1 (*Figure 3—figure supplement 2*). If *SIR4* is required for Ku's function in maintaining telomere length as a telomerase subunit, deleting *SIR4* should prevent TLC1 alleles with extra Ku-binding hairpins from causing telomere hyper-lengthening. We passaged *TLC1(Ku)$_3$* cells in liquid culture and assessed telomere length over time. TLC1(Ku)$_3$ caused progressive telomere hyper-lengthening over the course of passaging in addition to some telomere shortening (*Figure 3A*), similar to TLC1 RNAs with two Ku-binding sites (*Zappulla et al., 2011*). We also probed the Southern blot from *Figure 3A* for Y' telomeric restriction fragments and determined that telomeres in *TLC1(Ku)$_3$* cells range from ~70 bp shorter than wild type to ~1000 bp longer after 220 generations, continuing to progressively elongate at a rate of ~5 bp/generation (*Figure 3—figure supplement 3*). This increasingly heterogeneous distribution of telomere lengths in *TLC1(Ku)$_3$* cells could be due to diverse telomere lengths in the population of cells or an abnormality of telomeric DNA structure affecting how it migrates on gels (e.g., extremely long single-stranded tails). To differentiate between these possibilities, we plated the liquid culture-passaged cells for single colonies and found that telomeres from these clonal isolates were subsets of the heterogeneous liquid-cultured population (*Figure 3—figure supplement 1*), a behavior of

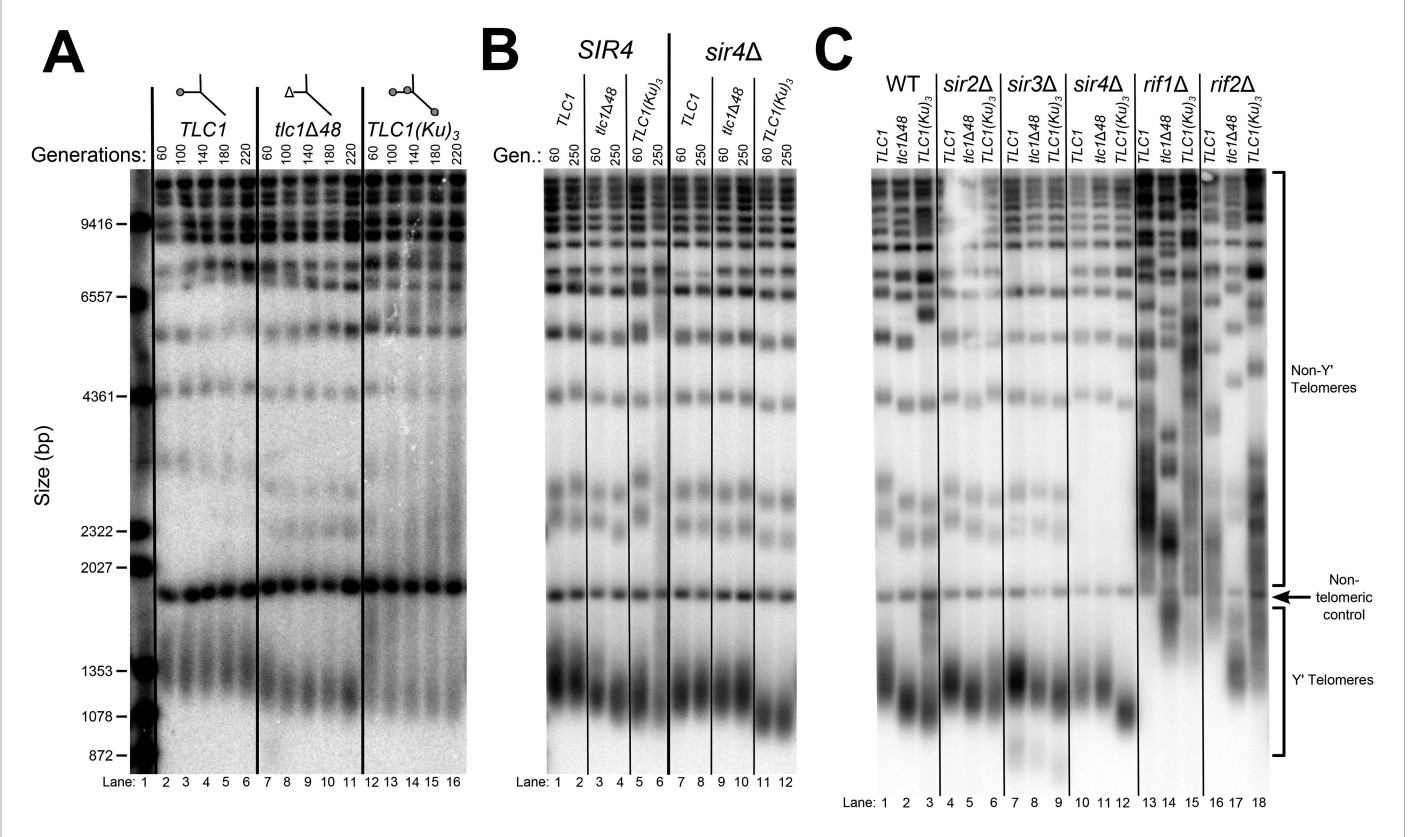

**Figure 3**. TLC1-bound Ku requires *SIR4* to promote telomere lengthening. (**A**) TLC1(Ku)₃, a TLC1 RNA containing two extra Ku-binding sites, causes both telomere hyper-lengthening and shortening. This experiment was performed as described in *Figure 2A*, except a *tlc1Δ pTLC1-LYS2 rad52Δ* strain was used. Additionally, instead of passaging cells on plates, single colonies were inoculated to liquid cultures, which were then serially passaged and harvested at various points throughout the passaging process. (**B**) TLC1(Ku)₃ does not cause telomere hyper-lengthening in *sir4Δ* cells. This experiment was performed as described in *Figure 2A*, but the liquid culture passaging method described in *Figure 3A* was used instead of re-streaking single colonies on plates. (**C**) TLC1(Ku)₃ does not cause telomere hyper-lengthening in *sir2Δ* or *sir3Δ* cells, and tlc1Δ48 causes greater telomere shortening in *rif1Δ* and *rif2Δ* cells than in wild-type cells. This experiment was performed as described in *Figure 3B* except that cells were passaged to ~250 generations by re-streaking on plates rather than passaging in liquid cultures.

The following figure supplements are available for figure 3:

**Figure supplement 1**. Three Ku-binding sites in yeast telomerase RNA increase telomere-length heterogeneity.

**Figure supplement 2**. TLC1 RNA abundance is largely unaffected in TLC1(Ku)₃ cells and is not decreased in *sir4Δ* cells.

**Figure supplement 3**. TLC1(Ku)₃ causes Y′-telomere shortening and hyper-lengthening, while deletion of *RIF1* or *RIF2* causes Y′-telomere hyper-lengthening.

telomeres that has been reported previously (*Shampay and Blackburn, 1988*; *Levy and Blackburn, 2004*). These results show that the wide variety in the relative mobility of telomeric restriction fragments in the gel is due to a broad distribution of telomere lengths from the population of cells.

Next, we tested if telomere hyper-lengthening caused by TLC1(Ku)₃ is dependent on *SIR4*. Whereas TLC1(Ku)₃ caused a combination of telomere hyper-lengthening and shortening in a wild-type *SIR4* strain, it did not cause any hyper-elongation in *sir4Δ* cells (*Figure 3B*). The average telomeres supported by TLC1(Ku)₃ in a *sir4Δ* strain were 148 ± 36 bp shorter than those in wild-type *TLC1 SIR4* cells (*Table 1*). This inability of TLC1(Ku)₃ to cause telomere hyper-lengthening without *SIR4* provides further evidence that Sir4 is required for telomerase RNA-bound Ku to promote telomere lengthening in yeast. We also tested whether the other two members of the Sir2/3/4 complex, Sir2 and Sir3, were

required for Ku-mediated telomere lengthening. We observed similar results in $sir2\Delta$ and $sir3\Delta$ cells, which, like $sir4\Delta$ cells, completely lack telomeric silencing, although telomeres supported by TLC1(Ku)$_3$ in these backgrounds were not quite as short as those supported by TLC1(Ku)$_3$ in a $sir4\Delta$ background (*Figure 3C*, *Table 1*). Of the three members of the Sir2/3/4 complex, only Sir4 has been identified as a binding partner for Ku by screening a two-hybrid library (*Tsukamoto et al., 1997*; *Roy et al., 2004*). Deleting *SIR2* or *SIR3* likely affects Ku-mediated telomere lengthening indirectly by substantially, but not completely, removing Sir4 from telomeres (*Hoppe et al., 2002* and see Figure 6).

In addition to promoting telomere lengthening, Ku binding to TLC1 is also known to increase telomerase RNA abundance (*Mozdy et al., 2008*; *Zappulla et al., 2011*). To test if the telomere-length phenotypes we observed are a function of RNA abundance, we assessed telomerase RNA levels by Northern blotting. We found that deleting the 48-nt Ku-binding site in TLC1 (tlc1$\Delta$48) reduced RNA abundance to 61 $\pm$ 22% the level of wild type, similar to what has been reported (*Figure 3—figure supplement 2*) (*Zappulla et al., 2011*). In contrast to TLC1, RNAs with two Ku-binding sites, which exhibit a ~20% increase in telomerase RNA abundance (*Zappulla et al., 2011*), TLC1(Ku)$_3$ showed little change relative to wild type (93 $\pm$ 9%). Although $sir4\Delta$ cells display a telomere-length defect very similar to $tlc1\Delta48$ cells, telomerase RNA abundance did not decrease in $sir4\Delta$ cells relative to wild type; in fact, it increased ~twofold in $sir4\Delta$ cells, and ~1.5-fold in $sir4\Delta$ $tlc1\Delta48$ cells, and remained near wild-type levels in $sir4\Delta$ TLC1(Ku)$_3$ cells. These results suggest that the telomere-length phenotypes shown in *Figure 3B* are not caused by decreased telomerase RNA abundance.

## Ku-mediated telomere lengthening is inhibited by Rif1 and Rif2

In the process of identifying proteins involved in Ku-mediated telomere lengthening, we tested the effects of tlc1$\Delta$48 and TLC1(Ku)$_3$ on telomere length in cells lacking Rif1 or Rif2, negative regulators of telomerase that bind to the same region of Rap1 as Sir3 and Sir4 (*Hardy et al., 1992*; *Wotton and Shore, 1997*; *Teng et al., 2000*). As shown previously, we found that both $rif1\Delta$ and $rif2\Delta$ cells have hyper-elongated telomeres (*Figure 3C* and *Figure 3—figure supplement 3*). Notably, we found that tlc1$\Delta$48 caused telomeres to shorten by ~500 bp in $rif1\Delta$ and $rif2\Delta$ cells. This is a substantially greater effect than the ~70-bp decrease caused by tlc1$\Delta$48 in a wild-type background, and it suggests that Rif1 and Rif2 inhibit Ku-mediated telomere lengthening. When we introduced TLC1(Ku)$_3$ into $rif1\Delta$ or $rif2\Delta$ cells, some telomeres became further hyper-elongated and others became shorter, suggesting that TLC1-bound Ku does not require Rif1 or Rif2 to promote telomere lengthening, in contrast to the requirement we identified for Sir4 as well as Sir2 and Sir3 shown in *Figure 3C*.

## Ku binds Sir4 in vitro

A binding interaction between Ku and Sir4 has been reported previously through yeast two-hybrid forward-genetic screens and by co-immunoprecipitation from yeast cell extracts (*Tsukamoto et al., 1997*; *Roy et al., 2004*). Using yeast two-hybrid, the N-terminus and C-terminus of Sir4 have been shown to interact with Yku80 and Yku70, respectively, and two different regions of Yku80 have been shown to be important for binding Sir4 (*Tsukamoto et al., 1997*; *Roy et al., 2004*; *Ribes-Zamora et al., 2007*). However, these studies do not rule out the possibility that Ku and Sir4 could be interacting indirectly, bridged by another yeast protein. Using purified yeast Ku heterodimer provided by the Cech lab (*Pfingsten et al., 2012*; *Dalby et al., 2013*), we tested for the Ku-Sir4 interaction in vitro. [$^{35}$S]-Sir4 was synthesized by using a rabbit reticulocyte lysate transcription/translation system (RRL) spiked with $^{35}$S-methionine. Prior to Sir4 protein synthesis, purified Ku heterodimer (Yku80-Myc•Yku70) was also added to the lysate. After Sir4 protein synthesis, Ku was then immunoprecipitated by anti-myc affinity pull-down. The input, unbound supernatant, and bound fraction were resolved on a gel and subjected to autoradiography. As shown in *Figure 4*, when Ku heterodimer was omitted from this procedure, a trace amount of radioactive Sir4 was recovered in the bound fraction, indicating a small amount of non-specific Sir4 binding the beads, and when Sir4 template DNA was omitted, no bands were detected. However, when both Ku and Sir4 template DNA were present in the RRL, ~fivefold more radioactive Sir4 was recovered in the bound fraction than in the no-Ku control, providing evidence for a direct interaction between the Ku heterodimer and Sir4. To test if this protein–protein interaction was specific, we repeated this experiment with Ku heterodimer that had been boiled before being added to the RRL. In this condition, only trace amounts of Sir4 were recovered in the bound fraction, similar to what was observed when Ku was

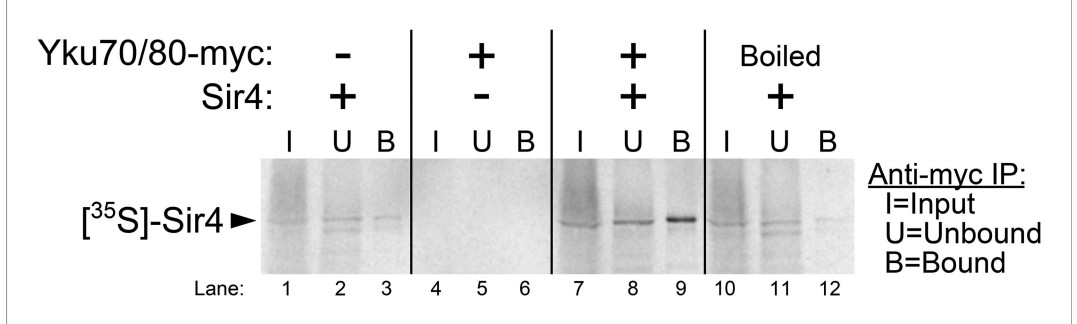

**Figure 4**. Purified Ku binds Sir4 in vitro. $^{35}$S-methionine-labeled Sir4 was synthesized in vitro in a rabbit reticulocyte lysate transcription/translation system (RRL) to which purified Ku heterodimer, bearing a 2myc epitope on the C-terminus of Yku80, was added. After Sir4 synthesis, the RRL was subjected to anti-myc immunoprecipitation. The input, unbound, and bound fractions were run on an SDS polyacrylamide gel, which was imaged by autoradiography.

omitted altogether. These results indicate that Sir4 participates in a specific protein–protein interaction with the Ku heterodimer, as has been suggested by previous in vivo experiments.

## *SIR4* is required for Ku-mediated telomerase recruitment to telomeres

Having shown that *SIR4* is required for TLC1-bound Ku to promote telomere lengthening, we next tested if *SIR4* is also required for Ku-mediated telomerase recruitment. We assessed the level of telomerase recruitment to telomeres in wild-type and sir4Δ cells expressing either TLC1, tlc1Δ48, or TLC1(Ku)$_3$ by performing ChIP on myc-tagged telomerase catalytic subunit, TERT (Est2). Quantitative real-time PCR (qPCR) was then used to determine the enrichment of a telomere-proximal locus relative to a telomere-distal control locus (*Sabourin et al., 2007*). We assayed for telomerase recruitment at telomeres VI-R and XV-L and observed highly similar results for both of these chromosome ends (*Figure 5A,B*). We observed that TERT enrichment at these telomeres in *tlc1Δ48* cells was reduced to 15% of wild type, similar to what has been reported previously (*Fisher et al., 2004*). In contrast, there was a ~10-fold increase in TERT at telomeres in *TLC1(Ku)$_3$* cells relative to wild type. However, in a sir4Δ background, enrichment of TERT at telomeres was decreased relative to wild type, regardless of which *TLC1* allele was expressed. The level of TERT at telomeres in *sir4Δ TLC1* and *sir4Δ tlc1Δ48* cells was reduced to 15% of wild type, and this is indistinguishable from what was observed in *SIR4 tlc1Δ48* cells (p = 0.80 and p = 0.91, respectively). In *sir4Δ TLC1(Ku)$_3$* cells, telomeric TERT enrichment was decreased to 35% the level of wild type, which is also similar to our observations in *SIR4 tlc1Δ48* cells (p = 0.11). These telomerase recruitment results provide molecular evidence that *SIR4* is required for Ku-mediated telomerase recruitment to telomeres.

We also performed ChIP for TERT in sir3Δ and sir2Δ backgrounds. Compared to wild-type cells, TERT enrichment at telomeres was reduced to 16% in *sir3Δ TLC1* cells, to 12% in *sir3Δ tlc1Δ48* cells, and to 39% in *sir3Δ TLC1(Ku)$_3$* cells (*Figure 5A*), similar to our observations in sir4Δ cells described above. TERT enrichment at telomeres was also decreased in a sir2Δ background relative to wild type but not as extensively as in a sir3Δ or sir4Δ background. In sir2Δ cells expressing TLC1, tlc1Δ48, or TLC1(Ku)$_3$, TERT enrichment at telomeres was reduced to 44%, 21%, or 86% of wild type, respectively. These data suggest that Sir3, and to a lesser degree Sir2, is also important for Ku-mediated telomerase recruitment to telomeres in addition to Sir4.

## Sir4 binding to telomeres is promoted by Sir2 and Sir3 and inhibited by Rif1 and Rif2

We have shown that Sir2 and Sir3 are important for Ku-mediated telomere lengthening and telomerase recruitment, but, unlike Sir4, neither Sir2 nor Sir3 has been shown to bind Ku. The simplest explanation is that Sir2 and Sir3 affect Ku-mediated telomerase recruitment indirectly through altering Sir4 association with telomeres, particularly since the amount of telomere-bound Sir4 has been shown

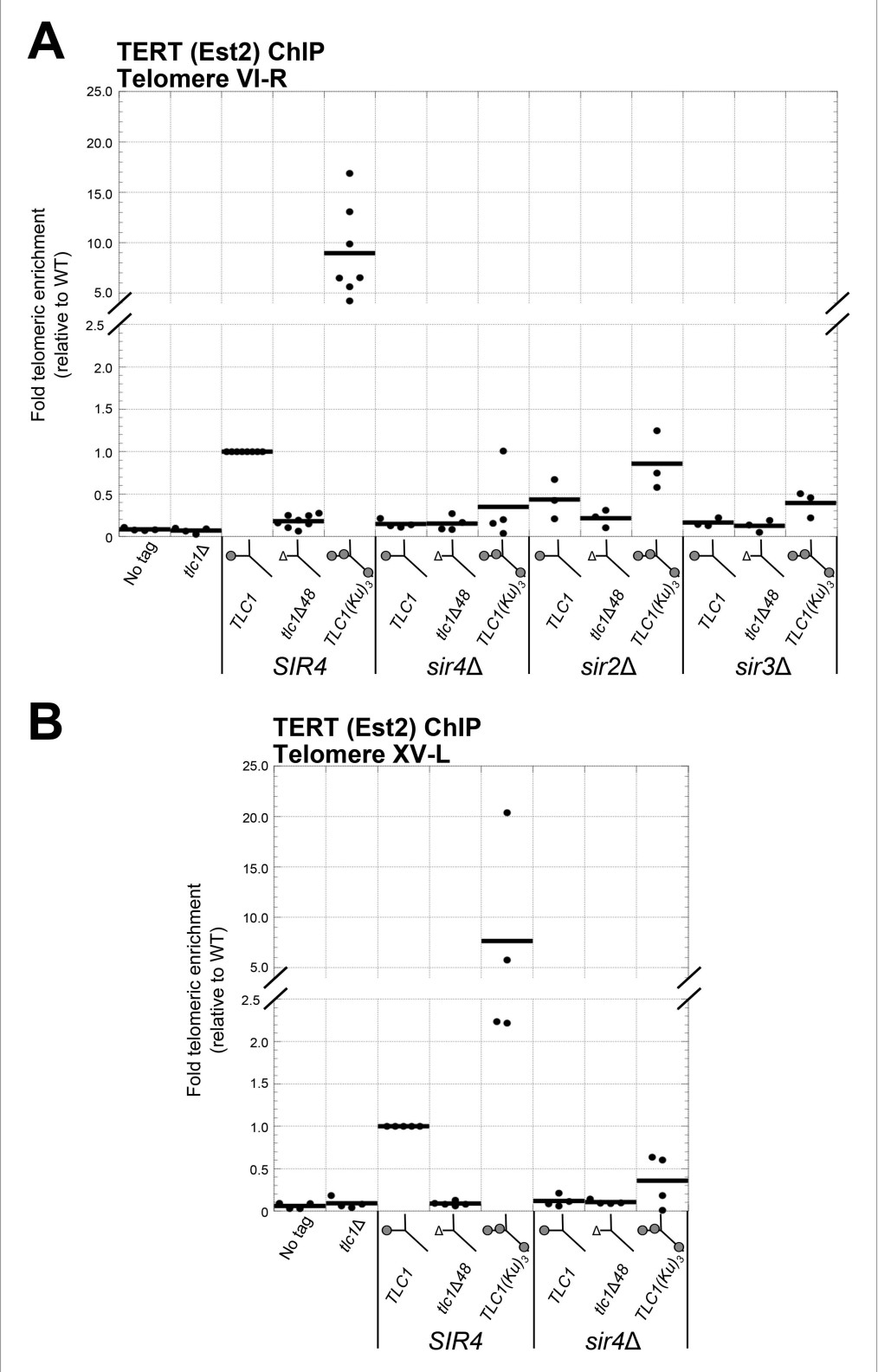

**Figure 5**. Ku-mediated telomerase recruitment to telomeres requires *SIR4*. (**A**, **B**) In strains similar to those used in *Figure 3B*, TERT (Est2) was expressed from its endogenous locus bearing a C-terminal 13myc tag, separated by an 8-glycine linker. TLC1, TLC1Δ48, and TLC1(Ku)₃ were expressed as in *Figure 3B*, but cells were not passaged after loss of the *pTLC1-URA3* cover plasmid. Cells were crosslinked and subjected to chromatin immunoprecipitation *Figure 5. continued on next page*

*Figure 5. Continued*

(ChIP) using the myc epitopes on TERT, as described (*Fisher et al., 2004*). Telomeric enrichment was measured using quantitative real-time PCR (qPCR) amplicons close to telomere VI-R (**A**) and telomere XV-L (**B**). An amplicon at the *ARO1* locus was used as a non-telomeric control locus. The thick horizontal lines on the graphs represent averages of three to five independent biological replicates, which themselves are indicated by black dots.

to decrease greatly, but not completely, in the absence of Sir2 or Sir3 (*Hoppe et al., 2002*). We have also shown that the proteins Rif1 and Rif2 function to inhibit Ku-mediated telomere lengthening. Because Rif1 and Rif2 compete with Sir3 and Sir4 for binding to Rap1 (*Moretti et al., 1994*; *Wotton and Shore, 1997*), this inhibition of Ku-mediated telomere lengthening could be explained by there being more Sir4 bound to telomeres in the absence of Rif1 or Rif2. To test this hypothesis, we performed ChIP on myc-tagged Sir4 in *rif1Δ* and *rif2Δ* cells as well as in *sir2Δ* and *sir3Δ* cells and used real-time quantitative PCR to measure fold telomeric enrichment. Similarly to what has been shown previously, we observed that Sir4 telomeric enrichment was decreased in *sir2Δ* and *sir3Δ* cells, to 26% and 17% of wild-type levels, respectively (*Figure 6*). In contrast, Sir4 telomeric enrichment was increased ~2.5-fold in *rif1Δ* cells and ~1.5-fold *rif2Δ* cells. These results suggest that Sir2, Sir3, Rif1, and Rif2 affect Ku-mediated telomerase recruitment and telomere lengthening by affecting the amount of Sir4 bound to telomeres.

### Tethering Sir4 to tlc1Δ48 RNA restores telomeres to wild-type length, while tethering Sir3 does not

Although disruption of the Est1-Cdc13 telomerase recruitment pathway results in an ever-shortening telomere phenotype, this can be rescued by tethering Cdc13 to telomerase through a protein fusion with TERT, effectively bypassing the need for Est1 in telomerase recruitment (*Evans and Lundblad, 1999*). Similarly, if Sir4 is in fact Ku's binding partner in telomerase recruitment, tethering Sir4 directly to telomerase RNA could rescue the short-telomere phenotype of *tlc1Δ48* cells. To test this, we tagged the sole chromosomal copy of *SIR4* with sequence encoding two tandem copies of the MS2 coat protein (MS2CP) and expressed either TLC1 or tlc1Δ48 with MS2 RNA hairpins, which have been employed previously to target proteins to telomerase RNA in yeast (*Gallardo et al., 2011*; *Lebo et al., 2015*). We passaged these cells for approximately 125 generations and then assessed telomere length.

As shown in *Figure 7*, tethering Sir4 to wild-type TLC1 resulted in telomeres ~30-bp longer than the wild-type no-tag control (compare lanes 22 and 23 to 2 and 3), and tethering Sir4 to tlc1Δ48 resulted in approximately wild-type length telomeres (lanes 24 and 25). To test the specificity of telomere length in *tlc1Δ48* cells being rescued by tethering Sir4 to the RNA, we also performed the same tethering experiment with Sir3, a telomeric silencing protein which has not been shown to bind Ku. When Sir3 was tethered to tlc1Δ48 (lanes 16 and 17), telomeres remained 107-bp shorter than the wild-type no-tag control. The MS2CP tags on Sir3 caused a small amount of telomere shortening (lanes 10 and 11), and tethering Sir3 to wild-type TLC1 (lanes 14 and 15) restored telomeres to approximately wild-type length, but, again, this was dependent on the 48-nt TLC1-binding site for Ku (lanes 16 and 17). In summary, the finding that specifically tethering Sir4 to Ku-binding-defective tlc1Δ48 RNA restores wild-type length telomeres provides direct support for Sir4 being the telomere-associated factor required for Ku-mediated telomerase recruitment.

### Discussion

Telomerase faces formidable challenges in binding and extending telomeres in the nucleus. Obstacles include the enzyme's extremely low concentration (*Mozdy and Cech, 2006*), the short period of time when telomerase has to act at the end of S phase (*Diede and Gottschling, 1999*), and the fact that chromosome ends are likely difficult to access due to telomeric heterochromatin (*Guertin and Lis, 2013*). Considering these impediments, it becomes clearer why telomerase would have multiple recruitment pathways assisting in providing enzyme access to telomeres. Furthermore, carefully regulating telomerase activity is critical – it is upregulated in 90% of cancers (*Shay and Bacchetti, 1997*) and reduced in telomere syndromes (*Armanios and Blackburn, 2012*) – and multiple pathways provide opportunities for layers of regulatory control. In *S. cerevisiae*, the primary

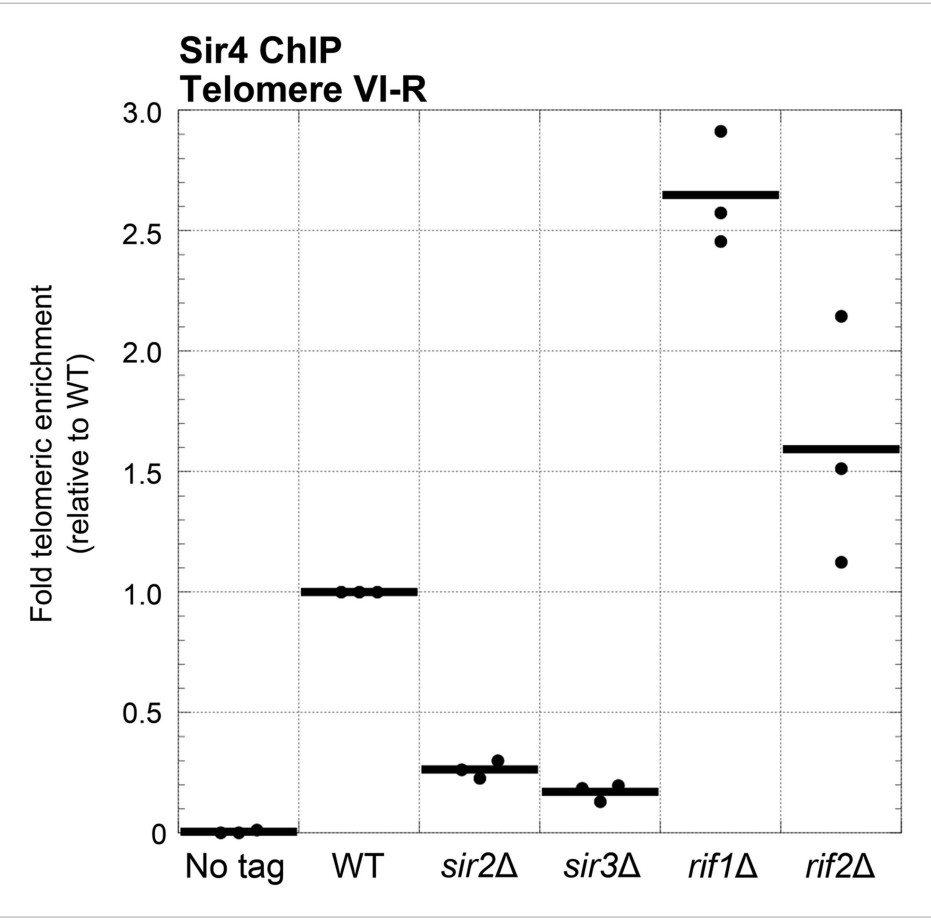

**Figure 6**. Sir4 binding to telomeres is decreased in *sir2Δ* and *sir3Δ* cells and increased in *rif1Δ* and *rif2Δ* cells. Sir4 bearing a C-terminal 13myc tag on an 8-glycine linker was expressed from its endogenous chromosomal gene locus. Cells were crosslinked and subjected to ChIP using the myc epitopes. Telomere VI-R enrichment was measured using real-time quantitative PCR as in *Figure 5A*. The thick horizontal lines on the graph represent averages of three independent biological replicates indicated by black dots.

telomerase recruitment pathway is essential and requires the Est1 telomerase subunit binding to the telomere-specific DNA-binding protein Cdc13 (*Evans and Lundblad, 1999*). In humans, it has been shown recently that telomere-bound Pot1•Tpp1 recruits telomerase through the Tpp1 'TEL patch' binding TERT (*Zhong et al., 2012*; *Nandakumar and Cech, 2013*; *Schmidt et al., 2014*). A second recruitment pathway in yeast has been identified that requires the Ku telomerase subunit and its binding to TLC1 (*Peterson et al., 2001*; *Stellwagen et al., 2003*; *Fisher et al., 2004*). In this study, we provide evidence that Ku recruits telomerase to telomeres in yeast through its binding to the telomeric transcriptional silencing protein Sir4.

There is substantial evidence for the existence of the Ku-mediated telomerase recruitment pathway in *S. cerevisiae*. Ku binding to TLC1 promotes telomere lengthening (*Peterson et al., 2001*; *Stellwagen et al., 2003*; *Zappulla et al., 2011*) and recruitment of the telomerase catalytic protein subunit to telomeres as assessed by ChIP (*Fisher et al., 2004*). Our current findings support this Ku-mediated recruitment pathway and provide the first evidence that it is achieved by binding the telomeric silencing protein Sir4. First, we report genetic epistasis between *SIR4* and the Ku-binding site in *TLC1* with respect to telomere-length maintenance. Second, we show that telomere hyper-elongation caused by a TLC1 RNA containing three Ku-binding sites, TLC1(Ku)$_3$, is dependent on *SIR4*. Third, using ChIP, we report that deleting *SIR4* causes low levels of telomerase catalytic subunit at telomeres and that this low degree of recruitment is indistinguishable from what is observed in *tlc1Δ48* cells. Furthermore, TLC1 with two extra Ku-binding sites causes a 10-fold increase in TERT

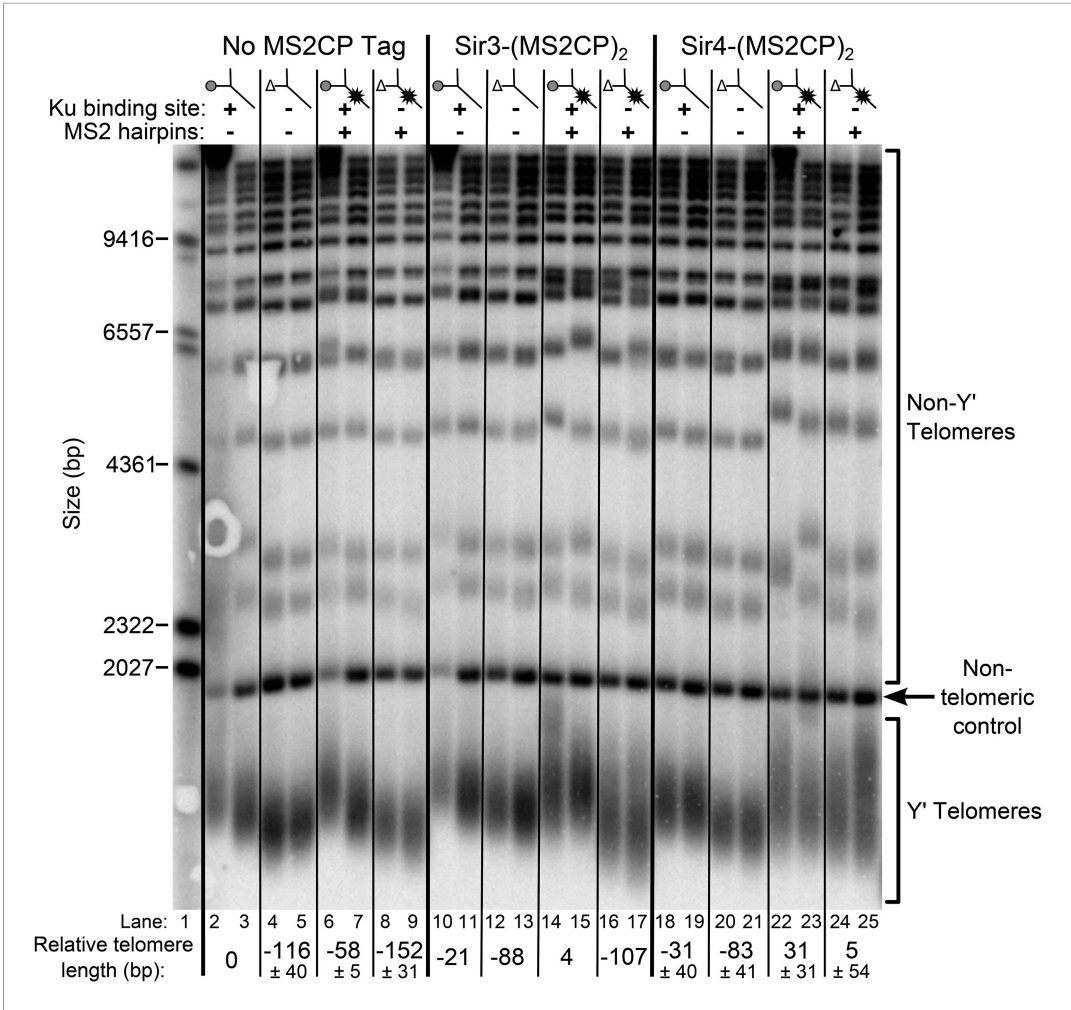

**Figure 7**. Tethering Sir4 to the tlc1Δ48 RNA restores telomeres to wild-type length. Using the same *tlc1Δ pTLC1-URA3* strain background from ***Figure 2A***, Sir3 and Sir4 were expressed from their endogenous loci bearing C-terminal (MS2CP)$_2$ tags, separated by an 8-glycine linker. These strains were transformed with *CEN* plasmids containing either *TLC1*, *tlc1Δ48*, *TLC1(MS2)$_{10}$*, or *tlc1Δ48(MS2)$_{10}$*. Cells were then cured of the *pTLC1-URA3* cover plasmid and passaged as in ***Figure 2A***. Each pair of lanes represents two independent biological replicates, and the relative telomere-length values are averages of the two replicates. In the no MS2 coat protein (MS2CP) tag and Sir4-(MS2CP)$_2$ conditions, values from a third set of replicates were included in the average, allowing for standard deviation to be calculated.

enrichment at telomeres, and this increase in telomerase recruitment requires *SIR4*. Finally, tethering Sir4 directly to a TLC1 RNA lacking its Ku-binding site restores telomeres to wild-type length, whereas tethering Sir3 to Ku-binding-defective TLC1 RNA does not. The fact that the loss of the telomere length-promoting function of the Ku-binding site in TLC1 can be rescued by directly tethering Sir4 to TLC1 provides strong evidence that Sir4 protein directly participates in Ku-mediated telomerase recruitment and telomere extension.

Based on the results presented here, we propose that Ku mediates telomerase recruitment to telomeres by binding to Sir4 (***Figure 8***). Since Ku-mediated telomerase recruitment occurs via Sir4 associating with telomeric DNA-bound Rap1, regulation of Sir4 association with Rap1, in turn, can control the ability of the Ku-Sir4 recruitment pathway to assist in telomere lengthening. As shown in ***Figure 6***, Sir4 association with telomeres is increased in the absence of Rif1 or Rif2, suggesting that Rif1 and Rif2 inhibit Sir4 binding to telomeres. The Rif1 and Rif2 proteins have been proposed to represent a 'counting mechanism' for telomere-length homeostasis in which decreased binding of the

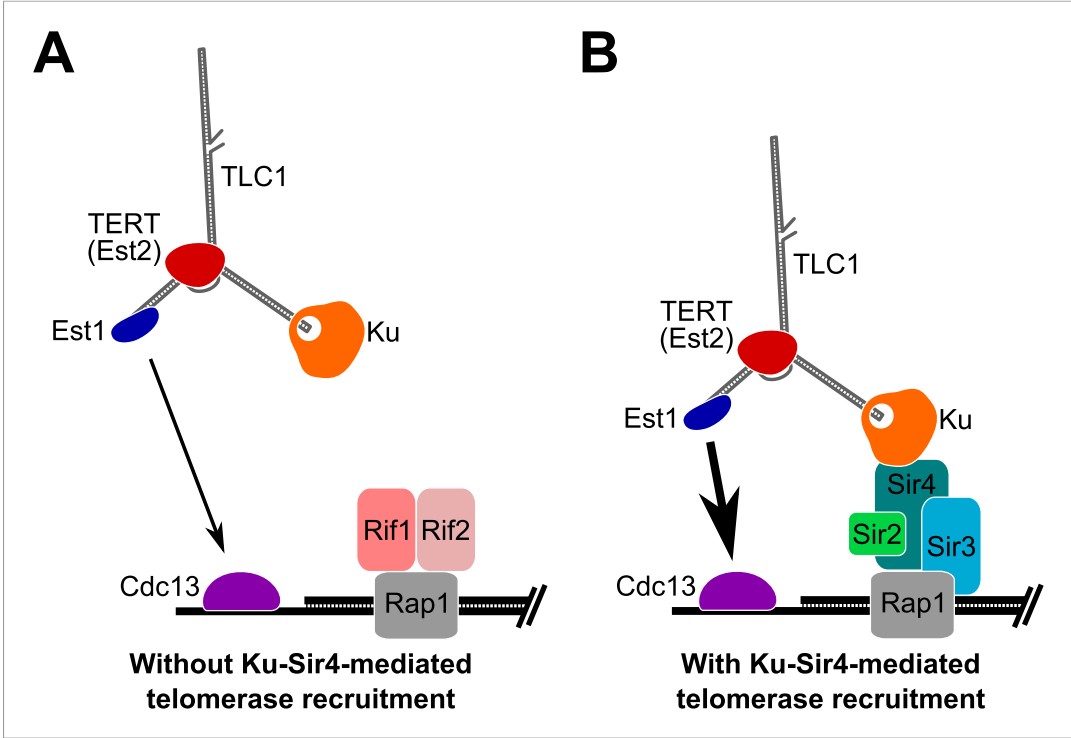

**Figure 8**. Model for Ku-Sir4 telomerase recruitment to telomeres and its role in telomere-length regulation in *Saccharomyces cerevisiae*. Telomerase has previously been shown to extend a telomere infrequently and shorter telomeres are preferentially extendable. We propose here that Ku recruits telomerase to telomeres by binding Sir4. Since it has been shown that Rif1 and Rif2 compete with Sir4 and Sir3 for binding telomere-bound Rap1, the Ku-Sir4 telomerase recruitment pathway is inhibited by Rif1 and 2, providing a simple mechanistic explanation for one way in which Rif proteins function to inhibit telomerase action at telomeres. (**A**) Ku recruitment of telomerase via Sir4 is inhibited by Rif1/2 competition for Rap1 binding with Sir4. In situations where Ku-Sir4-mediated telomerase recruitment does not occur, Est1-Cdc13-mediated telomerase recruitment can still happen, although with low efficiently. (**B**) When telomerase is recruited to a telomere through the Ku-Sir4 pathway, subsequent Est1-Cdc13-mediated recruitment to the end of the telomere becomes more efficient, resulting in increased telomerase extension of telomeres. The counterbalancing of Ku-Sir4 telomerase recruitment and Rif1/Rif2 occlusion of Sir4 binding to Rap1 may represent a system for maintaining telomere-length homeostasis in yeast.

Rif proteins – negative regulators of telomerase – leads to increased telomerase recruitment at short telomeres (*Marcand et al., 1997*; *Levy and Blackburn, 2004*; *Teixeira et al., 2004*; *Bianchi and Shore, 2007*). Our model provides a parsimonious explanation for one way in which Rif1 and Rif2 regulate telomere length; that is, as competitive inhibitors of Sir4 binding to Rap1, and therefore, the Sir4-Ku telomerase recruitment pathway. The model that the Ku-Sir4 recruitment pathway is subject to Rif protein-mediated negative regulation is supported by our findings and reports in the literature. First, we showed that TLC1(Ku)$_3$, a telomerase RNA with three Ku-binding sites, causes many telomeres to become hyper-elongated, which is similar to telomere hyper-elongation exhibited by *rif1Δ* and *rif2Δ* mutants. Second, deleting the Ku-binding site in TLC1 greatly reduces the telomere hyper-lengthening observed in *rif1Δ* and *rif2Δ* cells (see *Figure 3C*). Similarly, it has been shown that the hyper-lengthening of telomeres in *rif2Δ* as well as *rif1Δ rif2Δ* mutants is greatly reduced when combined with a *yku70Δ* mutation (*Mishra and Shore, 1999*). Regulation of the Ku-Sir4 pathway by Rif1 and Rif2 likely modulates, in turn, the essential Est1-Cdc13 telomerase-recruitment pathway. Ku-mediated telomerase recruitment has been proposed to promote function of the Est1-Cdc13 pathway by increasing Est1 association with telomeres (*Fisher et al., 2004*; *Williams et al., 2014*), a mechanism that is complementary to our model for Ku-mediated telomerase recruitment.

Sir4-mediated telomerase recruitment via Ku suggests that there is a relationship between (semi-stable) telomeric silencing and telomerase recruitment. Such a relationship has already been

suggested by the fact that Rif1 and Rif2 proteins compete with silencing proteins Sir3 and Sir4 for association with telomere-bound Rap1 and inhibit telomerase acting at longer telomeres. Accordingly, we propose that the Ku-Sir4 recruitment pathway tends to occur at shorter telomeres and is part of the negative-feedback loop regulating telomere length homeostasis. This is supported by telomerase preferentially acting at shortened telomeres (*Teixeira et al., 2004*), which tend to have weaker silencing than longer telomeres and have fewer Rif proteins (*Kyrion et al., 1993*; *Park and Lustig, 2000*). Our results and previous studies show that when telomeric transcriptional silencing is absent due to *sir2Δ* or *sir3Δ* mutation, a reduced but detectable amount of Sir4 is found at telomeres by ChIP (*Hoppe et al., 2002*). Thus, in wild-type cells, it may be that Sir4 binds to Rap1 to recruit telomerase via its Ku subunit without having established telomeric silent chromatin at the end. In summary, it seems most likely that short, non-silenced chromosome ends are the ones targeted for extension by the Sir4-Ku telomerase recruitment pathway and that this is an important part of the negative-feedback loop that maintains telomere-length homeostasis.

Although the Ku-Sir4 recruitment mechanism we propose is inhibited by the important negative regulators of telomerase Rif1 and Rif2, it is clear that the Ku-Sir4 pathway normally has a more modest role in telomere-length maintenance than the Est1-Cdc13 pathway. Disrupting the Ku-Sir4 pathway results in short but stable telomeres, whereas, in contrast, loss of Est1-Cdc13 recruitment causes complete loss of telomeres and cellular senescence. The typically smaller magnitude of the effects of the Ku-Sir4 pathway makes its disruption more likely to have been missed in prior studies. Furthermore, it has been difficult to separate the roles of Sir proteins in telomerase recruitment from competing negative roles of Rif1 and Rif2. The C-terminal domain of the Rap1 protein that binds telomeric dsDNA repeats is bound by Rif1 and Rif2 and recruits Sir3 and Sir4 to telomeres. Deleting the C-terminal domain of Rap1 therefore not only disrupts Sir-dependent silencing at telomeres but also abolishes Rif-protein inhibition of telomerase, and consequently, the net result is that telomeres become extremely long in a mutant lacking the Rap1 C-terminus (*Kyrion et al., 1992*). However, our model cannot fully explain the long-telomere phenotype of this mutant. In the absence of Rif1 and Rif2 binding at telomeres in *rap1Δc* cells, telomeres will only become hyper-elongated if a process that the Rif proteins inhibit persists. Because mutants lacking the Rap1 C-terminus have long telomeres despite having lost Sir4 binding at telomeres (at least via Rap1), the Ku-Sir4-mediated recruitment pathway we propose is not the only one inhibited by Rif1 and Rif2.

There are, however, a few noteworthy results in the literature suggesting that telomeric silent chromatin factors, particularly Rap1, contribute to telomere-length maintenance. First, deleting *SIR3* or *SIR4* causes telomere-length maintenance defects (*Palladino et al., 1993*; *Askree et al., 2004*; *Gatbonton et al., 2006*). Second, the Rap1 M763A mutation, which abolishes the Rap1-Sir3 interaction and slightly impairs telomeric silencing, has been shown to cause shortened telomeres (*Feeser and Wolberger, 2008*). Lastly, Rap1 binding near a telomeric seed sequence has been shown to promote telomerase-dependent de novo telomere formation, although this activity was reportedly *SIR4*-independent (*Ray and Runge, 1998*).

In summary, we have shown that telomerase RNA-bound Ku recruits telomerase to telomeres by binding the telomeric silent chromatin protein Sir4. The Ku-Sir4 pathway is inhibited by the telomerase regulators Rif1 and Rif2 and likely promotes telomerase recruitment through the essential Est1-Cdc13 recruitment pathway. Thus, this pathway represents an important mechanism by which telomerase is regulated to maintain telomere length in *S. cerevisiae* and it may be generally conserved in many other species. For instance, both telomeric silent chromatin and the Ku-telomerase RNA interaction are present in humans (*Baur et al., 2001*; *Ting et al., 2005*). Although humans lack an obvious Sir4 homolog, the protein HP1α has been implicated in human telomeric silencing (*Koering et al., 2002*; *Arnoult et al., 2012*) and has been shown to bind Ku70 (*Song et al., 2001*), so it will be interesting to learn if these interactions also comprise a telomerase-recruitment pathway.

## Materials and methods

### Experiments in yeast

Lists of the yeast strains and plasmids used can be found in *Supplementary file 1*. Experiments described in *Figures 2A, 3, 5, 7* are all based on *tlc1Δ* complementation assays reported previously (*Lebo et al., 2015*). Plasmid pRS414-based constructs containing *TLC1* alleles were transformed into

a *tlc1Δ* strain harboring a *pTLC1-LYS2-CEN* or *pTLC1-URA3-CEN* 'cover' plasmid. The *TLC1*-containing cover plasmid was then shuffled out by plating transformants on medium containing α-aminoadipate to select for LYS− cells that lost the *pTLC1-LYS2-CEN* cover plasmid or medium with 5-fluoroorotic acid to select for URA− cells that lost the *pTLC1-URA3-CEN* cover plasmid. In *Figure 5*, cells were streaked once to solid minimal medium lacking tryptophan before being grown for ChIP. In all other cases, cells were passaged by one of two methods after cover plasmid loss. In *Figures 2A, 7, 3C*, cells were passaged by serially re-streaking single colonies on solid minimal medium lacking tryptophan. When using this passaging technique, generation time was estimated as 25 generations per re-streak (including the streak used to shuffle out the cover plasmid). For *Figure 3A,B*, and *Figure 3—figure supplements 1, 2*, cells were passaged differently. First, after the streak used to shuffle-out the cover plasmid, cells were re-streaked once to solid minimal medium lacking tryptophan. Colonies from this minus-tryptophan (−TRP) plate were then used to inoculate 20-ml −TRP liquid cultures and culturing was performed at 30°C for ~24 hr. Cells were then back-diluted by a factor of $2^{10}$ into 20-ml cultures of fresh medium, which were grown for another ~24 hr before being passaged again. Cultures reached the same approximate density each day as measured spectrophotometrically (600-nm light). In these experiments, generation time was approximated as 50 generations (25 generations for the colony forming after streaking to solid medium when shuffling out the cover plasmid plus 25 more for the −TRP medium growth) plus 10 generations for each day of passaging in liquid cultures. In *Figure 2B*, a *SIR4/sir4Δ YKU80/yku80Δ* diploid was sporulated, and tetrads were dissected to isolate tetratype spores. The spores from this ascus were then re-streaked three successive times on rich YPD medium before telomere length was assessed.

## Southern blotting

Southern blotting was performed as described previously (*Zappulla et al., 2005*, *2011*). Briefly, cells were pelleted either directly from liquid cultures used for passaging or from cultures grown from serial re-streaking plates. Genomic DNA was isolated from these cells (Gentra Puregene system from Qiagen, Hilden, Germany), and roughly equal amounts of genomic DNA were digested with XhoI. Digested genomic DNA samples were resolved on a 1.1% agarose gel. The DNA was then transferred to Hybond-N+ Nylon membrane (GE, Little Chalfont, United Kingdom), which was probed for telomeric sequence and for a 1627-bp, non-telomeric XhoI restriction fragment from within chromosome IV and then imaged using phosphor screens and a Typhoon 9410 Variable Mode Imager (GE) (*Friedman and Cech, 1999*). Average Y′ telomere length was calculated using the weighted average mobility method as previously described (*Zappulla et al., 2011*). In *Figure 3—figure supplement 3*, Southern blots were probed for Y′ sequence. Y′ probe was made by first performing PCR with the following primers using genomic DNA as template DNA: 5′-TGTTGTCTCTTACCCGGATGTTCAACC-3′, 5′-AAAGTTGGAGTTTTTCAGCGTTTGCG-3′. The DNA amplified in this reaction was in turn used as template for making the radiolabeled Y′ probe.

## Northern blotting

Northern blotting was performed as previously described (*Zappulla et al., 2005*). Briefly, cells were harvested in the same manner as those used for Southern blots, and total RNA was isolated using the hot-phenol method (*Kohrer and Domdey, 1991*). 10–15 μg of RNA from each sample was boiled and then resolved by urea-PAGE. The RNA was transferred to Hybond-N+ Nylon membrane (GE), which was then UV-crosslinked and probed for TLC1 and U1 sequences. Due to the low abundance of TLC1 RNA relative to U1, blots were probed with 100-fold fewer counts of U1 probe than TLC1 probe. Blots were then imaged using phosphor screens and a Typhoon 9410 Variable Mode Imager (GE).

## ChIP and quantitative PCR

ChIP was performed similarly to that described (*Fisher et al., 2004*). Briefly, cells were grown to saturation in 10-ml cultures of liquid minimal medium, back-diluted into 60 ml cultures, and then grown to an OD600 of 0.5–0.8. 50 ml of cells were crosslinked with formaldehyde, pelleted, rinsed in lysis buffer, and then re-suspended in lysis buffer. Cells were flash-frozen, thawed, and then lysed using sterile glass beads. Lysates were then sonicated to shear crosslinked chromatin. Anti-myc immunoprecipitation was carried out using mouse anti-myc monoclonal antibodies (Clontech, Mountain View, CA, United States) and Protein G

Dynabeads (Life Technologies Oslo, Norway). After immunoprecipitation, formaldehyde crosslinks were reversed, and DNA was purified using reagents from the Qiagen PCR Purification Kit.

Fold telomeric enrichment in ChIP DNA samples was quantified by qPCR using iQ SYBR Green Supermix and a CFX96 Real–Time Cycler (Bio-Rad Hercules, CA, United States). The primer sets used at telomere VI-R, telomere XV-L, and the *ARO1* locus were the same as those described previously (*Sabourin et al., 2007*; *McGee et al., 2010*). For a given sample of DNA obtained from ChIP, qPCR reactions for each primer set were performed in technical duplicate or triplicate, and the $C_T$ values were averaged together. Using these averages, fold telomeric enrichment was then calculated as $2^{[(C_{T(ARO\ IP)} - C_{T(ARO\ Input)}) - (C_{T(TEL\ IP)} - C_{T(TEL\ Input)})]}$. Additionally, each time qPCR was performed, the efficiency of amplification was calculated for each primer set being used. From a sample of ChIP input DNA, a series of 10-fold dilutions were made and used as template DNA for qPCR reactions. For these reactions, $-\log(\text{dilution factor})$ was plotted against $C_T$ value, and a line of best fit was found for the graph. Using the slope of this line, percent amplification efficiency was calculated as $100*[10^{(-1/slope)} - 1]$. If amplification efficiency was between 70% and 95%, average $C_T$ values were corrected using the slope and Y-intercept values from the line of best fit: Relative amount = $10^{[(AvgC_T - intercept)/slope]}$. Then, fold telomeric enrichment was instead calculated as $(\text{RelAmt}_{TEL\ IP}/\text{RelAmt}_{TEL\ Input})/(\text{RelAmt}_{ARO\ IP}/\text{RelAmt}_{ARO\ Input})$. Fold telomeric enrichment values in *Figure 5* are expressed relative to wild type (*SIR4 TLC1*).

### In vitro protein–protein binding experiments

First, ~16 pmol of purified, myc-tagged yeast Ku heterodimer (*Pfingsten et al., 2012*; *Dalby et al., 2013*) was added to the RRL transcription and translation system (TNT Quick Coupled, Promega Madison, WI, United States). In the 'boiled' condition in *Figure 4*, the Ku heterodimer was heated at 95°C for 5 min before being added to the RRL. Sir4 synthesis was then initiated by adding 1 µg of *SIR4* template DNA (plasmid pDZ930) and $^{35}$S-L-methionine to the RRL, and the reaction was incubated at 30°C for approximately 90 min. 5 µl of mouse anti-myc monoclonal antibodies (Clontech, used at a 1:400 dilution in TBST) was added to the reaction, which was then incubated at 4°C for 1 hr. 40 µl of Protein G Dynabeads (Life Technologies) was prepared for each RRL reaction by first pipetting off the storage buffer, rinsing once in 1 ml of 'standard' Ku-Sir4 buffer (25 mM HEPES pH 7.5, 100 mM NaCl, 1 mM DTT, 10% glycerol, 1 mM EDTA, 0.1% IGEPAL), and then re-suspended in 40 µl of 'standard' Ku-Sir4 buffer per RRL reaction. Before adding beads to the RRL reactions, a 2 µl aliquot was taken from the RRL and set aside to be used as the input sample for the protein gel. 40 µl of prepared beads was added to each RRL reaction, and the reactions were left to rotate at 4°C overnight. The next morning, the beads were pulled down with a magnet, and a 2 µl aliquot of the supernatant was set aside to be used as the unbound sample for the protein gel. The remaining supernatant was discarded, and the beads were washed twice with 500 µl of 'stringent' Ku-Sir4 buffer (same as 'standard' Ku-Sir4 buffer but with 200 mM NaCl and 0.2% IGEPAL). Beads were then re-suspended in 120 µl TE +1% SDS and heated at 95°C for 5 min. The beads were pulled down with a magnet, and the entire supernatant was saved as the bound fraction. 2 µl of 2× protein sample buffer was added to the input and unbound aliquots, which were then heated at 95°C for 5 min. These samples along with a bound sample (10 µl of the bound fraction plus 10 µl of 2× protein sample buffer) were resolved by SDS-PAGE on a 7.5% polyacrylamide gel. The resulting gel was imaged using phosphor screens and a Typhoon 9410 Variable Mode Imager. It should be noted that the 'standard' and 'stringent' Ku-Sir4 buffers were designed based off of the buffers used in co-immunoprecipitation experiments described previously (*Roy et al., 2004*).

## Acknowledgements

We thank Robert Schleif, Haiqing Zhao, and members of the Zappulla laboratory for comments on the manuscript, Tom Cech's laboratory for the generous gift of purified Ku heterodimer, and Virginia Zakian's laboratory for advice with ChIP and quantitative PCR procedures. This research was supported by a March of Dimes Basil O'Connor Starter Scholar Research Award and National Institutes of Health (National Institute of General Medical Sciences) funding from R00 GM80400 to DCZ, in addition to startup funds from The Johns Hopkins University. EPH has been supported by National Institutes of Health Cellular and Molecular Biology graduate student training grant 2T32 GM007231.

## Additional information

### Funding

| Funder | Grant reference | Author |
|---|---|---|
| March of Dimes Foundation | #5-FY12-91 | David C Zappulla |
| National Institute of General Medical Sciences (NIGMS) | GM080400 | David C Zappulla |

The funders had no role in study design, data collection and interpretation, or the decision to submit the work for publication.

### Author contributions

EPH, Conception and design, Acquisition of data, Analysis and interpretation of data, Drafting or revising the article; DCZ, Conception and design, Analysis and interpretation of data, Drafting or revising the article

### Author ORCIDs

David C Zappulla, http://orcid.org/0000-0001-8242-3493

## Additional files

### Supplementary file
• Supplementary file 1. (**A**) Yeast strains used in this study. (**B**) Plasmids used in this study.

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
