## [Decision Letter]

Thank you for submitting your work entitled “The Ku subunit of telomerase binds Sir4 to recruit telomerase to lengthen telomeres in *S. cerevisiae*” for peer review at *eLife*. Your submission has been favorably evaluated by James Manley (Senior editor) and two reviewers, one of whom is a member of our Board of Reviewing Editors. The other review, David Shore, has agreed to share his identity.

The reviewers have discussed their comments with one another and the Reviewing editor has drafted this decision to help you prepare a revised submission.

Essential revisions:

1) As Sir4 can bind telomeres in the absence of other Sir proteins, the authors need to address why *sir2* and *sir3* mutants have short telomeres (the smaller amount of Sir4 bound not sufficient?). Most important, in my opinion, is that they measure Sir4 ChIP in the *rif* mutants, but they should also do this in *sir2* and *sir3*, as well as to measure Est2 (TERT) in all the *sir* mutants.

2) All reviewer comments must be addressed. Some other experiments would be nice, but are not required.

Comments 1 and part of 3 from Reviewer 1 are the main ones.

Reviewer #1:

This paper provides evidence that Ku promotes telomere lengthening via a direct interaction with Sir4. Although the Ku-Sir4 interaction has been previously described (2-hybrid), and Sir4 and Ku are known to be important for normal telomere length, the mechanistic connection of the Ku-Sir4 interaction to telomere length is unknown and important. As such, I think the paper is appropriate in principle for publication in *eLife*. However, there are some unexplained aspects of the model and a few missing experiments that need to be performed prior to acceptance.

1) The crux of the paper hinges on the specific interaction of Ku and Sir4. In accord with this, the authors show that Sir4 is required for recruitment of TERT (Est2) to telomeres. However, it is essential to analyze TERT recruitment in Sir2 and Sir3 mutant strains. To go along with this, it will also be important to analyze Sir4 occupancy in the mutant strains.

2) The Sir4 tethering experiment is very nice. However, it is unclear why Sir3 tethering doesn't work, because it is my understanding that all the Sir proteins are bound to the telomere. In this case, why does it matter which protein is tethered? This needs to be analyzed by Sir2, Sir3, and Sir4 ChIP in the relevant strains.

3) Rif1 and Rif2 are clearly important in the story and model, but the experiments here are limited. Sir protein ChIP experiments need to be done in the *rif1*/*rif2* mutant strains carrying the relevant alleles of telomerase RNA. In addition, Rif1 and Rif2 ChIP experiments in the tethered strain and the various telomerase RNA alleles also need to be done.

4) In the ideal world, the authors would conclusively prove their model by obtaining derivatives of Sir4 and/or Ku that selectively block their interaction without affecting other functions of these proteins. Such derivatives should affect telomere length without affecting silencing or other aspects of telomere function.

Reviewer #2:

In this manuscript Hass and Zappulla identify a new telomerase recruitment pathway in the budding yeast *S. cerevisiae*. Previous work had implicated the yeast Ku heterodimer (yKu) in telomerase recruitment through a well-characterized interaction with the telomerase RNA template (TLC1). Thus, yKu bound to telomeric DNA was thought to directly recruit telomerase. However, a more recent study from the Cech lab showed that yKu could not simultaneously bind DNA and RNA in vitro, putting this simple model into question. Hass and Zappulla present a convincing case, based on genetic, biochemical and ChIP data that telomere-bound Sir4 protein instead binds to yKu, which in turn recruits telomerase through the previously described yKu-TLC1 interaction. Although this pathway does not substitute for the essential and more thoroughly studied Cdc13-Est1 telomerase recruitment pathway, the authors argue that this additional pathway may improve the robustness of the system for telomerase extension and provide additional means for its regulation.

I believe that the findings reported here are novel and significant, and will be of considerable interest both to researchers in the telomere field and to a wider audience. I have the following specific comments:

1) Prior evidence for the Sir4-yKu interaction is not clearly described and the evidence presented here is relatively weak (Figure 4). Has a domain of Sir4 been identified that is sufficient? Can the interaction be measured with purified proteins?

2) The observation (Figure 3) that the *TLC1(Ku)*_*3*_ mutant appears to “cause telomeric restriction fragments to be progressively more heterogeneous over time” does not seem to me to be a very good description of the data. To me it seems that a population of very short Y' telomeres is maintained whereas the rest of the Y' telomeres elongate progressively over time. Analysis of clones from these cultures (Figure 3—figure supplement 1) doesn't clarify this issue, and I wonder if it might be useful to probe the blots only for Y' telomeres. The authors claim that a similar observation has been made in two other publications; details should be provided and discussed. Since there are cases where extensive telomere elongation is observed in the absence of this apparently bi-modal distribution, the authors should at least discuss this issue and propose a testable model to explain the observation. I did not understand their comments on this (subsection “Telomere hyper-lengthening by telomerase RNA with extra Ku-binding sites is *SIR4*-dependent”).

3) The observation regarding the effect of *tlc1Δ48* in cells lacking either Rif1 or Rif2 is interesting (Figure 3) and should be looked at more closely. Again, it might be useful to specifically examine Y' ends, since the heterogeneity of non-Y' telomeres and their frequent co-migration with the Y' ones complicates the analysis. The authors should test whether *sir4Δ* causes a further reduction (or not, as they would presumably predict). Again, the effect of TLC1(Ku)_3_ in these mutants was complex, with some telomere shortening evident. This needs to be addressed.

---

## [Author Response]

*1) As Sir4* can *bind telomeres in the absence of other Sir proteins, the authors need to address why* sir2 *and* sir3 *mutants have short telomeres (the smaller amount of Sir4 bound not sufficient?). Most important, in my opinion, is that they measure Sir4 ChIP in the* rif *mutants, but they should also do this in* sir2 *and* sir3*, as well as to measure Est2 (TERT) in all the* sir *mutants*.

We constructed 7 new strains and have now performed all of these requested ChIP experiments. The current Figure 6 is new and shows Sir4 ChIP in wild-type, *rif1*∆, *rif2*∆, *sir3*∆ and *sir2*∆ strains. We have also completed Est2 ChIP in all *sir* mutants, as requested, and this is shown in Figure 5.

The results of all of these experiments provide additional strong evidence supporting our proposed Sir4-Ku telomerase recruitment model and its relationship with the Rif-based counting model for telomere-length regulation (see Figure 8). We agree with the reviewers that shorter telomeres in *sir2* and *sir3* mutants is explainable due to reduced Sir4 at telomeres in these strains, as we now show this to be the case from the new results in Figure 6, which were what we expected based on earlier data from Danesh Moazed’s lab (25). The reduced Sir4 occupancy of telomeres in *sir3*∆ and *sir2*∆ strains could simply be mechanistically explained by the fact that Sir2/3/4 form a heterotrimeric complex and when one of the factors is absent, the trimer cannot form appropriately and this reduces the formation of higher-order complexes that normally enhance Sir4 recruitment (in addition to being a trimer, Sir3 can bind itself and Sir4, Sir4 can bind itself and Sir3, and Sir2 binds Sir4). It is possible that Sir4 molecules that are part of a higher-order Sir2/3/4 multimeric complex (i.e., Sir4 not directly bound to Rap1) can assist in telomerase recruitment and/or it could be that Sir4 bound to Rap1 is available for the Ku interaction and that the absence of Sir2 or Sir3 actually affects telomere-association of the complex due to altering Sir4’s ability to associate with Rap1 as effectively as in wild-type cells.

The requested experiment examining Sir4 ChIP in *rif1*∆ and *rif2*∆ mutants shows that Sir4 occupancy at telomeres is increased, consistent with these factors acting to inhibit Sir4 at telomeres from the “counting model” originally proposed by the Shore and Blackburn labs and our model for this functioning via the Sir4-Ku telomerase-recruitment pathway. We appreciate the reviewers suggesting this experiment, as it adds new results for the field and additional evidence for the Sir4-Ku recruitment model.

The new results showing Est2 ChIP in *sir3*∆ and *sir2*∆ mutants (Figure 5) correlate nicely with the added Sir4 ChIP data in Figure 6. Deleting *SIR3* causes a slightly more profound effect on telomerase recruitment than deleting *SIR2*.

*2) All reviewer comments must be addressed. Some other experiments would be nice, but are not required*.

We have addressed all of the reviewers’ comments below, in many cases adding experimental results and/or adding or editing the manuscript text.

Reviewer #1:

[…] However, there are some unexplained aspects of the model and a few missing experiments that need to be performed prior to acceptance.

*1) The crux of the paper hinges on the specific interaction of Ku and Sir4. In accord with this, the authors show that Sir4 is required for recruitment of TERT (Est2) to telomeres. However, it is essential to analyze TERT recruitment in Sir2 and Sir3 mutant strains. To go along with this, it will also be important to analyze Sir4 occupancy in the mutant strains*.

We have performed all of the experiments requested by Reviewer 1, as described above in our response to the essential revisions; we did ChIP for TERT in *sir2*∆ and *sir3*∆ strains and Sir4 ChIP as well in *sir2*∆, *sir3*∆, *rif1*∆ and *rif2*∆ strains. The results were those expected if the Sir4-Ku model for telomerase recruitment is valid as well as its relationship to the telomere length-regulating counting model.

We agree with the reviewer that specific interaction of Ku and Sir4 is important for the paper. This is a reason why we did the in vitro-binding studies shown in Figure 4 to reconstitute this interaction and show that it is direct.

Ku and Sir4 are not quite required for recruitment of TERT to telomeres (they are not essential to avoid senescence), but these factors are clearly important for the majority of recruitment as assessed by TERT ChIP. Our data confirm that *sir4*∆ and Ku-binding-defective TLC1 (*tlc1∆48*) have very similar deficits in telomerase recruitment, as would be expected if Sir4 is required for the Ku telomerase subunit to recruit the enzyme to telomeres.

*2) The Sir4 tethering experiment is very nice. However, it is unclear why Sir3 tethering doesn't work, because it is my understanding that all the Sir proteins are bound to the telomere. In this case, why does it matter which protein is tethered? This needs to be analyzed by Sir2, Sir3, and Sir4 ChIP in the relevant strains*.

As requested and as described above in the response to the essential revisions, we have now also done Sir4 ChIP, in wild-type, *sir2*∆, *sir3*∆, *rif1*∆ and *rif2*∆ strains that we constructed, with expected outcomes. Reviewer 1 points out here that our results show that the ability to productively recruit telomerase is specific to Sir4. We are also interested in why it is that Sir4 is the specific Sir protein that has evolved to be involved in Ku-mediated telomerase recruitment. This topic is the basis for future studies as it will involve a significant effort to evaluate potential hypotheses for why this is the case. It could be an issue of position of Sir4 along the telomere or subtelomere, or an issue of timing in the cell cycle as Sir proteins (individual molecules vs subcomplexes) are recruited to Rap1, etc.

*3) Rif1 and Rif2 are clearly important in the story and model, but the experiments here are limited. Sir protein ChIP experiments need to be done in the* rif1*/*rif2 *mutant strains carrying the relevant alleles of telomerase RNA. In addition, Rif1 and Rif2 ChIP experiments in the tethered strain and the various telomerase RNA alleles also need to be done*.

As Sir4 ChIP was part of the requested essential revisions, which we performed in multiple strains including *rif1*∆ and *rif2*∆ as requested here, this request has now been addressed above and the new data are shown in Figure 6. The newly added results are described in the revised Results and Discussion sections. These results expand on the Rif1 and Rif2 aspects of this paper and further validate their role in the proposed model. In the two months suggested for revisions and as they were not part of the essential revisions, we could not also address the Rif1 and Rif2 ChIP requests in the variety of telomerase RNA conditions.

*4) In the ideal world, the authors would conclusively prove their model by obtaining derivatives of Sir4 and/or Ku that selectively block their interaction without affecting other functions of these proteins. Such derivatives should affect telomere length without affecting silencing or other aspects of telomere function*.

We agree with the reviewer that this set of experiments would be great. We have concerns, however, that it may not be possible to obtain such findings, as the Ku interaction with Sir4 is important not only for telomerase recruitment via Ku as a telomerase subunit, but also for Sir2/3/4 recruitment to telomeres, which in turn affects the former process. Presumably the same Ku-Sir4 interaction interface is used for both functions and since Ku’s central role in recruitment of Sir2/3/4 to telomeres is “upstream” of its other role in recruiting telomerase as a subunit of the RNP to telomere-associated Sir4. This is a very interesting dual role of Ku, but it will be the subject of future studies to examine it and understand why telomere biology involves this multifaceted Ku functionality. Perhaps it is an important facet of the telomere length-regulating negative feedback loop.

Reviewer #2:

*[…] I believe that the findings reported here are novel and significant, and will be of considerable interest both to researchers in the telomere field and to a wider audience. I have the following specific comments*:

*1) Prior evidence for the Sir4-yKu interaction is not clearly described and the evidence presented here is relatively weak (*Figure 4*). Has a domain of Sir4 been identified that is sufficient? Can the interaction be measured with purified proteins?*

We have added to the description details of the interaction and referencing of the prior evidence for the Sir4-Ku heterodimer association in the subsection “Ku binds Sir4 in vitro” of the revised manuscript to give a better sense of what is known about these questions and what is newly contributed by this publication.

*2) The observation (*Figure 3*) that the* TLC1(Ku)_3_
*mutant appears to “cause telomeric restriction fragments to be progressively more heterogeneous over time” does not seem to me to be a very good description of the data. To me it seems that a population of very short Y' telomeres is maintained whereas the rest of the Y' telomeres elongate progressively over time. Analysis of clones from these cultures (*Figure 3—figure supplement 1*) doesn't clarify this issue, and I wonder if it might be useful to probe the blots only for Y' telomeres. The authors claim that a similar observation has been made in two other publications; details should be provided and discussed. Since there are cases where extensive telomere elongation is observed in the absence of this apparently bi-modal distribution, the authors should at least discuss this issue and propose a testable model to explain the observation. I did not understand their comments on this (subsection “Telomere hyper-lengthening by telomerase RNA with extra Ku-binding sites is* SIR4*-dependent”)*.

We appreciate the reviewer suggesting Y′ probing of our Southern blots to add specificity about the elongation vs shortening within this population of 17 of the total 32 telomeres in yeast. We have performed Y′ probing on the blots from Figure 3 and show these in a new Figure 3—figure supplement 3.

We also appreciate the reviewer pointing out that our description of the telomere length heterogeneity could be more accurate, so we revised the relevant section substantially to try to be clearer. The Y′ probing data in the new figure supplement is similar to what we saw with telomeric repeat-probing in Figure 3 for these telomeres. It does generally seem that there are two qualitatively different subpopulations of telomeres; those that are hyperelongating and those that are shortened and certainly it is not a simple Gaussian distribution spread over a wider length range. However, it is not obvious even from Y′ probing that the hyperelongating telomeres are a subgroup that behaves uniformly; they seem rather heterogeneously spread of a range of lengths. We have adjusted the text, including the lines referred to above, to help reflect the entirety of our results.

The Y′ probing has also been helpful in that it allows us to get a more quantitative sense of telomere length hyperelongation extent (∼1000 bp) and the rate (∼5 bp/generation) for the elongating subpopulation in the *TLC1(Ku)*_*3*_ cells. This information is referred to in the text.

*3) The observation regarding the effect of* tlc1Δ48 *in cells lacking either Rif1 or Rif2 is interesting (*Figure 3*) and should be looked at more closely. Again, it might be useful to specifically examine Y' ends, since the heterogeneity of non-Y' telomeres and their frequent co-migration with the Y' ones complicates the analysis. The authors should test whether* sir4Δ *causes a further reduction (or not, as they would presumably predict). Again, the effect of TLC1(Ku)*_*3*_
*in these mutants was complex, with some telomere shortening evident. This needs to be addressed*.

As mentioned in the point above, we have now probed some of the key blots for Y′ telomeres. The results of this probing show that what we presumed were Y′ telomeres in telomeric repeat-probing data in Figure 3 are indeed Y′ ends. With respect to the suggestion of the epistasis experiment (deleting *SIR4* to test for lack of a further reduction in *rif*∆ *tlc1*∆*48* cells, etc), this is a relevant, interesting experiment, but we were not able to add it to the substantial experimentation, involving several newly constructed strains, triplicate ChIP experiments for Sir4 and Est2, and Y′ Southern blots, accomplished in the two months suggested. This was also due to the fact that strain constructions for the particular set of TLC1-allele testing in novel triple–mutant combinations proposed would be significantly involved due to unavailability of markers in the strains.